# Data-driven discovery of electrocatalysts for $CO_2$ reduction using active motifs-based machine learning

Dong Hyeon Mok[1,3], Hong Li[2,3], Guiru Zhang[2], Chaehyeon Lee[1], Kun Jiang [2] ✉ & Seoin Back [1] ✉

The electrochemical carbon dioxide reduction reaction ($CO_2$RR) is an attractive approach for mitigating $CO_2$ emissions and generating value-added products. Consequently, discovery of promising $CO_2$RR catalysts has become a crucial task, and machine learning (ML) has been utilized to accelerate catalyst discovery. However, current ML approaches are limited to exploring narrow chemical spaces and provide only fragmentary catalytic activity, even though $CO_2$RR produces various chemicals. Here, by merging pre-developed ML model and a $CO_2$RR selectivity map, we establish high-throughput virtual screening strategy to suggest active and selective catalysts for $CO_2$RR without being limited to a database. Further, this strategy can provide guidance on stoichiometry and morphology of the catalyst to researchers. We predict the activity and selectivity of 465 metallic catalysts toward four expected reaction products. During this process, we discover previously unreported and promising behavior of Cu-Ga and Cu-Pd alloys. These findings are then validated through experimental methods.

Global $CO_2$ emissions from fossil fuels combustion and industrial processes have rapidly increased and reached their highest-ever annual levels every year. To reduce the atmospheric concentration of $CO_2$, there has been growing interest in renewable energy-driven electrochemical $CO_2$ reduction reaction ($CO_2$RR) as a means of upgrading $CO_2$[1]. Recently, several studies have been conducted to screen active, selective, and cost-efficient $CO_2$RR catalysts[2–5].

Introduction of energetic descriptors using scaling relations between binding energy of reaction intermediates and machine learning (ML) methods has been especially helpful in exploring large chemical spaces and greatly accelerating the discovery of high-performing catalysts. This approach has helped overcome the high computational costs associated with the complex reaction pathways of $CO_2$ reduction and the time-consuming density functional theory (DFT) calculations required for evaluating too many candidate materials[6–8]. For example, Tran and Ulissi developed an automated

screening method for discovering bimetallic $CO_2$RR catalysts using regression ML models and CO* binding energy as an activity descriptor[9]. Using this approach, Cu-Al catalyst was proposed to be active for $CO_2$RR, and experimental validations demonstrated a high faradaic efficiency toward ethylene production, confirming the validity of the catalyst discovery approach using both DFT and ML[2]. However, it should be noted that in screening studies, CO* binding energy was used as the activity descriptor to predict only the catalytic activity of $CO_2$RR, without considering the selectivity of products. Since a variety of $CO_2$RR products distribution have been experimentally reported[10], it is also crucial to evaluate the selectivity of products. Further, most screening research relies on open databases such as the Materials Project, which limits the chemical space to explore and makes it difficult to discover entirely new catalysts.

As an alternative approach to property prediction, inverse design has become a promising method for predicting materials with desired

[1]Department of Chemical and Biomolecular Engineering, Institute of Emergent Materials, Sogang University, Seoul 04107, Republic of Korea. [2]Interdisciplinary Research Center, School of Mechanical Engineering, Shanghai Jiao Tong University, Shanghai 200240, China. [3]These authors contributed equally: Dong Hyeon Mok, Hong Li. ✉e-mail: kunjiang@sjtu.edu.cn; seoin0226@gmail.com

target properties[11]. By using generative models, one can encode high-dimensional chemical space of materials into the continuous latent space of low dimensionality, and generate new materials using the knowledge embedded in the latent space[12,13]. As a similar methodology to the inverse design, a data-driven and machine-learning-enabled high throughput virtual screening (HTVS) strategy also has been applied to explore a vast chemical space. The HTVS strategy involves the initial creation of materials pool followed by property prediction. Although technically different from the inverse design, which generates materials with the specific properties, HTVS can fulfill the same essential objective of identifying materials with desired properties within an unknown chemical space without the need for time-consuming steps, as long as the materials pool is sufficiently extensive[14]. We previously developed a DFT and structure-free active motif-based representation (DSTAR) of catalyst surfaces for binding energy prediction, which can be used in HTVS[7]. With this method, one can enumerate all possible active motifs for any elemental combinations and construct histograms of predicted binding energies.

In this work, we developed a workflow by combining binding energy prediction ML models based on DSTAR and $CO_2RR$ selectivity map to discover active and selective catalysts (Fig. 1a). Based on the concept of $CO_2RR$ selectivity map originally developed by Tang et al.[15], we used three binding energies to more accurately predict the activity and selectivity. Using this method, we evaluated the potential-dependent activity and selectivity of $CO_2RR$ for 465 binary combinations without performing any DFT calculations and surface structure modeling. We further demonstrated that our method is capable of providing more detailed design strategy by analyzing the activity and selectivity according to composition and coordination number of active motifs. Finally, we experimentally validated Cu-Pd and Cu-Ga binary alloys and confirmed their high selectivity for $C_{1+}$ and formate, respectively, which agreed with our prediction by HTVS. We expect that the HTVS strategy developed in this work will accelerate the discovery of active and selective $CO_2RR$ catalysts.

## Results

### Enumeration of active motifs

The advantage of DSTAR workflow is that one can expand the chemical space to explore by numerically substituting elements of fingerprints. The DSTAR method includes the positional information of active motifs by dividing them into three sites: the first nearest neighbor (FNN) atoms of the adsorbates, the second nearest neighbor atoms in the same layer ($SNN_{same}$), and the sublayer of the binding site ($SNN_{sub}$) (Fig. S1). The active motif representation in DSTAR does not require time-consuming steps such as slab structure generations, binding site identifications and iterative optimizations, allowing for the exploration of a wide chemical space, thus facilitating a discovery of novel catalysts[9,11]. The more detailed explanation about DSTAR can be found in Supplementary Note 1. Based on this workflow, we collected 5634 (408) unique active motifs of bimetallic (monometallic) surfaces based on CO* data in GASpy dataset, which contains 89 types of crystal structures[9]. The active motifs were substituted by 30 monometallic and 435 bimetallic combinations consisting of 30 elements (Fig. S2). This resulted in an increase in the number of bulk structures from 1,089 in GASpy to 279,690. All surface structures in dataset and corresponding information can be found in https://github.com/SeoinBack/DSTAR-CO2RR. Note that DSTAR is not limited to binary compositions but can be expanded to more diverse compositions. However, we only focused on pure metals and binary alloys in this work for simplicity and abundance of training data. As a result, we generated a total of 2,463,030 active motifs and predicted their $\Delta E_{CO^*}$, $\Delta E_{OH^*}$ and $\Delta E_{H^*}$ using the trained ML models. Figure 1b demonstrates parity plots of DFT calculated and ML predicted binding energies with their test MAEs of 0.118, 0.227 and 0.107 eV for $\Delta E_{CO^*}$, $\Delta E_{OH^*}$ and $\Delta E_{H^*}$, respectively, based on fivefold cross validation (Table S1). The achieved

accuracies, although slightly lower compared to those of state-of-the-art ML models based on crystal graphs such as LS-CGCNN[8], can still be considered reasonable, as the decrease in accuracy is not substantial. Moreover, it is important to note that our goal is to explore a large chemical space without performing time-consuming steps. Therefore, we opted for a simpler ML model, DSTAR, that employs elemental descriptors of the nearest neighbors. Even at the cost of reduced accuracy, this model enables HTVS through the expansion of the chemical space, an aspect that is limited for complex neural networks that require precise geometric information as input and extensive modeling of surface structures. Additionally, we note that certain elements caused significantly higher MAEs in predictions compared to the overall MAEs (Fig. S3). We confirmed that those elements were not included in screening dataset.

### Potential-dependent 3D selectivity map for $CO_2RR$

It has been reported that binding energies of various intermediates of $CO_2RR$ (Fig. S4) can be estimated by scaling relations using $\Delta E_{CO^*}$ and $\Delta E_{OH^*}$[16–18]. It allows to establish thermodynamic boundary conditions, constructing a selectivity map that predicts $CO_2RR$ products. Although there are multiple reaction pathways possible for $C_1$ products and beyond[10], we focused on 7 reactions and 6 thermodynamic boundary conditions to predict the selectivity of 4 main products of $CO_2RR$, i.e., formate, CO, $C_{1+}$ (>2e⁻) and $H_2$, as previously reported by Tang et al.[15], where $C_{1+}$ (>2e⁻) corresponds to further reduced products than CO*/CO (g) (Table 1). In summary for each boundary conditions, boundary condition (1) ($BC_1$) compares reaction energy between two initial protonation steps, reaction (i) and reaction (ii), to determine whether the reaction pathway leads to formate or CO / $C_{1+}$ pathway. $BC_2$ evaluates the favorability of the Volmer step, while $BC_3$ considers the possibility of surface poisoning by OH*. The $BC_4$ ensures the binding strength of CO* for further CO* reduction and determines whether the product will be CO (g) or $C_{1+}$. The $BC_5$ considers competition between HER and $CO_2RR$, and the $BC_6$ assesses the favorability of the Heyrovsky reaction. The detailed descriptions, assumptions and derivations of reaction mechanisms and boundary conditions can be found in Supplementary Note 2. The scaling relation to derive the boundary conditions can be found in Fig. S5. Note that we used the directly predicted $\Delta E_{H^*}$ in the analysis to avoid multiple uncertainties originated from ML and scaling relations as shown in Fig. S6 (0.218 eV and 0.107 eV of MAE for using scaling relation and direct prediction, respectively), while the original study by Tang et al.[15] estimated $\Delta E_{H^*}$ using the scaling relation between $\Delta E_{H^*}$ and $\Delta E_{CO^*}$. Thus, we presented the selectivity map in three dimensions using three binding energy descriptors ($\Delta E_{CO^*}$, $\Delta E_{H^*}$ and $\Delta E_{OH^*}$ as x, y and z-axis, respectively) (Fig. 2).

We validated the constructed selectivity map by comparing the predicted results with experimental observations. Three binding energies on face-centered cubic (FCC) (111) and (211) facets of pure metals were calculated and plotted on the selectivity map (Fig. 2). In the following, we discuss four typical classes of catalysts, confirming that 3D selectivity map well captures characteristics of catalysts[19].

(i) The late transition metals, such as Rh, Ir and Pt, were reported to be selective for a competitive $H_2$ evolution reaction (HER). This is due to strong binding strengths of both CO* and H*, locating them on $H_2$ selective region (purple)[20–22]. Note that Pd is the exception of this group producing CO mainly, because Pd easily transforms into hydrides ($PdH_x$) making CO* binding energy considerably weaker[23–25]. Weakened CO* binding then shifts $PdH_x$ to CO selective region (red).

(ii) The coinage metals, such as Au and Ag, favor CO production[26–28] since their $\Delta E_{OH^*}$ and $\Delta E_{CO^*}$ are weak, preventing the formation of HCOO* (BC. 1) and further protonation of CO* to form COH* (BC. 4), respectively.

(iii) The p-block elements are reported to be selective for formate production[29]. For example, Pb is located in formate selective region (green), whose $\Delta E_{CO^*}$ is weaker than $\Delta E_{OH^*}$ making it more selective for

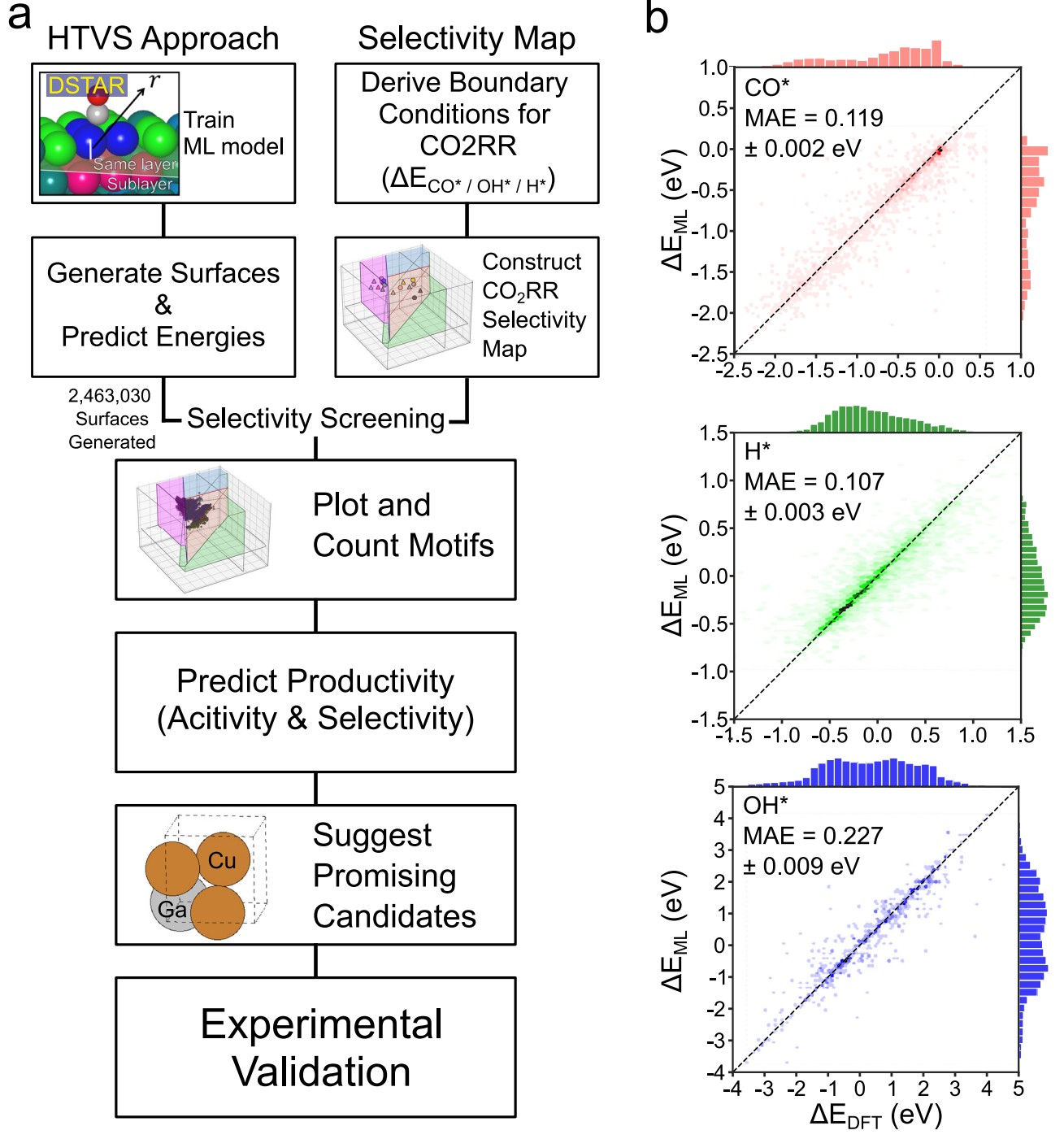

**Fig. 1 | High-throughput virtual screening schematics and performance of ML model. a** Schematics of the high-throughput virtual screening strategy to discover selective $CO_2RR$ catalysts. **b** Parity plots of ML-predicted ($\Delta E_{ML}$) and DFT-calculated ($\Delta E_{DFT}$) binding energy of CO* (red), H* (green) and OH* (blue). The prediction mean absolute error (MAE) values were calculated by averaging MAEs of fivefold splits (Table S1).

HCOO* formation than COOH*, leading to the sole production of formate (BC. 1).

(iv) Cu is an unique catalyst producing various products due to moderate $\Delta E_{CO*}$, $\Delta E_{OH*}$ and $\Delta E_{H*}$[30,31]. Both Cu (111) and Cu (211) are located in $C_{1+}$ selective region (blue).

### Activity and selectivity prediction of binary alloys

Since the 3D selectivity map in Fig. 2 successfully explained the selectivity toward $CO_2RR$ products, we then use this map to predict the selectivity and activity of binary alloy catalysts using their ML-predicted $\Delta E_{CO*}$, $\Delta E_{OH*}$ and $\Delta E_{H*}$, as shown in Fig. 3a. From the plotted points of the active motifs on the selectivity map, we identified the expected product using boundary conditions, and calculated the activity towards the expected products using the Gibbs free energy diagram derived from the binding energy scaling relations. Given that each elemental combination consists of 5643 unique active motifs, encompassing various geometries of bulk crystal structures and their corresponding surfaces obtained from the GASpy database, the comprehensive evaluation of overall selectivity and activity trends requires the consideration of the contributions of each active motif and the prediction errors of the ML model. To address this, we have introduced a new metric referred to as "productivity" in this work. This metric

serves as an indicator of both activity and selectivity at the applied potential. We emphasize that productivity provides a quantitative representation of both activity and selectivity within a single value. This differs from most previous HTVS studies for $CO_2RR$, which primarily focused on predicting activity alone by solely calculating $\Delta E_{CO^*}$. Moreover, productivity considered the ML prediction error by incorporating a term of probability that the active motif is positioned within the range of errors, covering the uncertainty of predicted binding energy. Additionally, since the productivity value comprehensively characterizes the catalyst by incorporating thousands of data points, it helps mitigate the uncertainty arising from the discontinuity of boundary conditions. The details about calculating productivity can be found in method section and Fig. S7. The top 20 candidates promising for each product and their corresponding productivity are enumerated in Table S2.

Figure 3b illustrates the productivity of four $CO_2RR$ products for 30 pure metals and 435 binary alloys at U = −1.4 $V_{RHE}$. The productivity heatmap is dependent on the applied potential and can demonstrate potential dependency of activity and selectivity, because each productivity is also dependent on the applied potential. For example, Fig. S8, productivity heatmap at U = −1.0 $V_{RHE}$, shows that selectivity of Cu is shifted from formate to $C_{1+}$ as more negative potential is applied

in agreement with literature[32]. Notably, $CO_2RR$ electrocatalytic behaviors of most of the alloy systems in Fig. 3b have not been reported in literature yet (Fig. S9), even though the experimentally well-developed systems generally agree with our productivity prediction results, e.g., a classification of metal electrodes into four groups suggested by Hori et al.[19]; formate (Pb, Hg, Tl, In, Sn, Cd, and Bi), CO (Au, Ag, Zn, Pd and Ga), $H_2$ (Ni, Fe, Pt and Ti) and further reduced products such as $CH_4$ and $CH_3OH$ (Cu). We note, however, that a few discrepancies were observed. Ag is expected to be selective towards formate at the applied potentials more positive than −1.3 $V_{RHE}$ (Fig. S10), while it is reported to primarily produce CO and $H_2$. This disagreement, also raised by Tang et al.[15], can be attributed to extrinsic factors such as kinetics[33,34] and local field effects[35], which are not included in the selectivity map constructed based on the thermodynamic boundary conditions. More advanced descriptors that employ constant potential DFT method and proton transfer barrier calculations should be developed to tackle this problem, which will be discussed in the follow-up study. Further, the selectivity map predicts Ga and Zn to be formate-selective, while there is no consensus in the experiments. Hori et al. suggested Ga and Zn to be CO-selective[19], while more recent studies found that they are formate-selective[36,37]. More thorough experiments would be of help for clarity.

## Composition and coordination number-dependent productivity analysis

Since the ML models used in this work are based on DSTAR representation, one can extract composition and coordination number (CN) information of the active motifs to investigate their effects on the productivity (Fig. S12). Particularly, the ratio of facet sites with high CN to edge/corner sites with low CN in nanoparticle catalysts is determined by the size and shape of catalysts, thus selectively masking composition or CN of the active motifs could help to understand the productivity trends[28,38]. We selected Cu-Al alloys to discuss CN and composition-dependent productivity in the following.

Figure 4a, b demonstrated $C_{1+}$ productivity of Cu-Al catalysts with respect to Al contents and CN, respectively, at −1.4 $V_{RHE}$. Figure 4c illustrated potential-dependent changes in $C_{1+}$ productivity when various conditions of active motifs are applied. We found increasing $C_{1+}$ productivity with decreasing Al contents and CN. This suggests

### Table 1 | Reaction steps and boundary conditions

| Reactions | | Boundary Conditions (BC) | |
|---|---|---|---|
| $CO_2(g) + H^+ + e^- + * \rightarrow COOH^*$ | (i) | $\Delta G_{rxn}^{(i)} = \Delta G_{rxn}^{(ii)}$ | (1) |
| $CO_2(g) + H^* \rightarrow HCOO^*$ | (ii) | $\Delta G_{rxn}^{(iii)} = 0$ | (2) |
| $H^+ + e^- + * \rightarrow H^*$ | (iii) | $\Delta G_{rxn}^{(iv)} = 0$ | (3) |
| $OH^* + H^+ + e^- \rightarrow H_2O + *$ | (iv) | $\Delta G_{rxn}^{(v)} = 0.75$ | (4) |
| $CO^* + H^+ + e^- \rightarrow COH^* (G_{CO^*} < 0)$ $CO(g) + H^+ + e^- + * \rightarrow COH^* (G_{CO^*} > 0)$ | (v) | $\Delta G_{rxn}^{(vi)} = 0$ | (5) |
| $CH^* + H^* \rightarrow CH_2^* + *$ | (vi) | $\Delta G_{rxn}^{(vii)} = 0$ | (6) |
| $H^* + H^+ + e^- \rightarrow H_2 + *$ | (vii) | | |

The reaction steps in $CO_2RR$ and thermodynamic boundary conditions to construct 3D selectivity map. $\Delta G_{rxn}$ is the reaction Gibbs free energy. More details can be found in Supplementary Note 2.

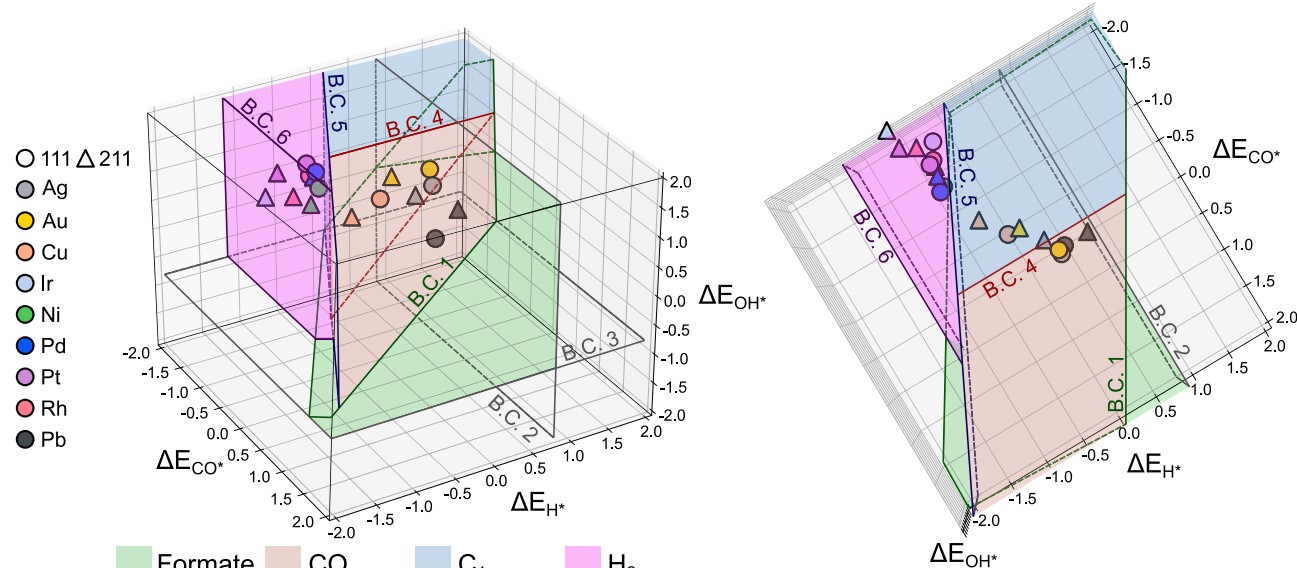

**Formate**  **CO**  **$C_{1+}$**  **$H_2$**

**Fig. 2 | 3D selectivity map.** Two perspectives of potential-dependent 3D selectivity map for $CO_2RR$ using three binding energy descriptors ($\Delta E_{CO^*}$, $\Delta E_{OH^*}$ and $\Delta E_{H^*}$) at U = −1.0 $V_{RHE}$. Green, red, blue and purple correspond to formate, CO, $C_{1+}$ and $H_2$ selective regions, respectively. No products are expected beyond outskirts of the map, since neither $CO_2RR$ nor HER is energetically favorable at the applied potential.

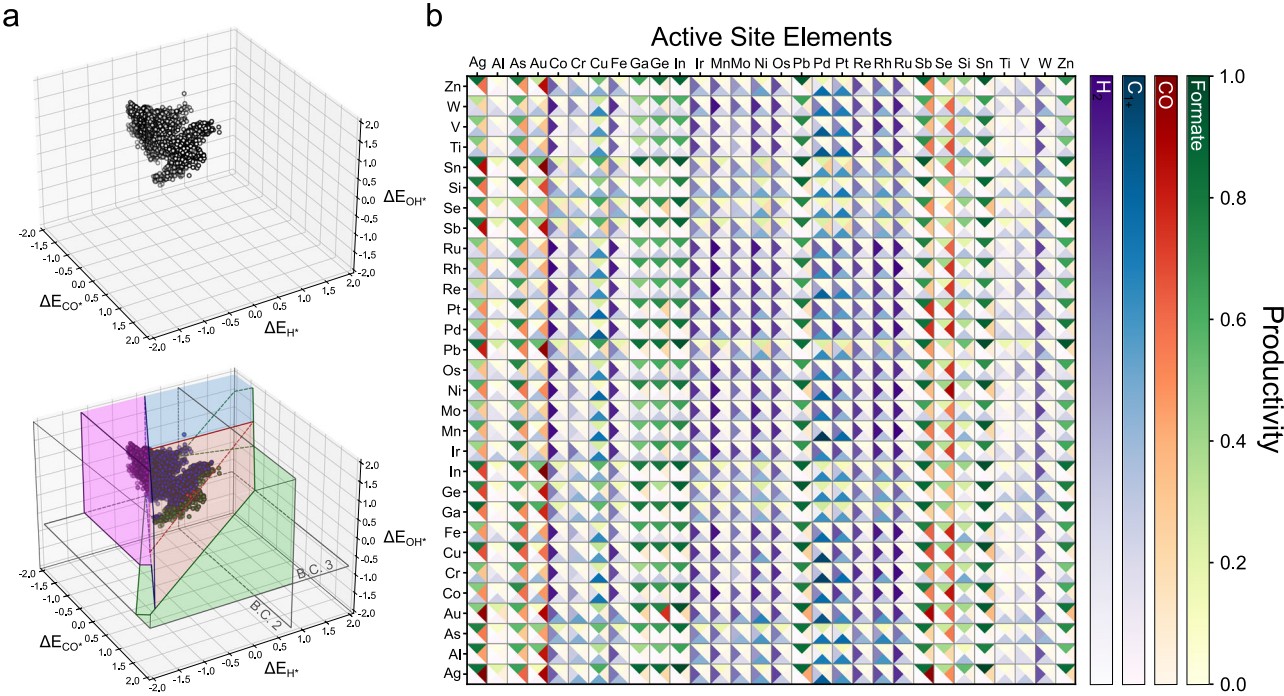

**Fig. 3 | Visualization of prediction method and results. a** Visualization of the 3D selectivity map with plotted points representing unique active motifs, and the process of identifying the expected product using the boundary conditions.

**b** Productivity heatmap visualizing the normalized productivity; Formate (green, top), CO (red, right), $C_{1+}$ (blue, bottom), $H_2$ (purple, left) at U = −1.4 $V_{RHE}$. The metal elements in y-axis are active metal sites on which adsorbates bind.

that the productivity can be tuned by controlling the compositions and/or the morphologies of catalysts. For example, if $C_{1+}$ is the target product, low Al content and high portion of edge/corner sites with low CN would be a suitable catalyst design principle. Although Cu-Al is selective for $C_{1+}$ products, its $C_{1+}$ productivity is lower than that of pure Cu when no conditions are applied (Fig. 4c, d). However, when the condition of low Al content and low CN is applied, its productivity significantly improves over pure Cu (Fig. 4d). This illustrates that, despite the improvement in $C_{1+}$ productivity with decreasing Al content in the Cu-Al alloy (Fig. 4a), the addition of Al to Cu is still predicted to be more beneficial for $C_{1+}$ production than not adding any at all. This behavior is in agreement with previous reports, which demonstrated the $AlCu_3$ (211) exhibited the highest $CO_2RR$ activity and selectivity compared to pure Cu, $Al_2Cu$ and $AlCu_3$ (111) surfaces[2]. In short, ML-assisted HTVS strategy provides important guidelines to optimize catalysts for the maximized target productivity by fine-tuning compositions, sizes and shapes of catalysts.

## Catalysts discovery

To evaluate the reliability and discoverability of the HTVS method, we focused on two aspects of catalyst discovery: promise and novelty. (1) A promising catalyst is predicted to have extraordinary activity, and (2) a novel catalyst is predicted to demonstrate rarely reported or entirely new catalytic behavior. As an example of a promising catalyst, we found that the Cu-Pd alloy is predicted to be the most active and selective towards $C_{1+}$ production, boasting the highest $C_{1+}$ productivity among 465 candidates. Additionally, we discovered that the Cu-Ga alloy ranks in the top 10 % for formate productivity at −1.4 $V_{RHE}$, a notable result since previous reports about the Cu-Ga catalyst focused on selectivity towards $C_{1+}$ and $C_{2+}$ products[3]. In the following sections, we will discuss the ML-predicted results and provide experimental validations of our ML-assisted HTVS strategy.

As shown in Fig. 5a, ML-predicted productivities of Cu-Pd and Cu-Ga demonstrate that $C_{1+}$ and formate production are dominant at negative potential, respectively. However, while $C_{1+}$ productivity of

Cu-Pd steadily increases as the potential increases in the negative direction, reaching a maximum value, the formate productivity of Cu-Ga monotonically decreases after −1.4 $V_{RHE}$. Methodologically, this trend for Cu-Ga is caused by the expansion of the $C_{1+}$ selective area. This is because the potential term of boundary condition 1 (BC 1), which determines the selectivity between $C_{1+}$ and formate (Table S3), is linearly proportional to $\Delta E_{OH^*}$ (0.601 $\Delta E_{CO^*}$ − 0.740 $\Delta E_{OH^*}$ + $\Delta E_{H^*}$ + 1.343 + eU = 0). As the applied potential decreases, the $C_{1+}$ selective area above the plane of the boundary condition increasingly takes up the area of formate (Fig. S13). Consequently, formate productivity decreases while $C_{1+}$ productivity increases. In aspect of electrochemistry, we compared the $CO_2RR$ performance on electro-plated Cu-Ga versus that on polished Cu in 0.1 M $CO_2$-saturated $CsHCO_3$ electrolyte (Supplementary Note 3, Figs. S14–S20). Similar steady-state current densities were noted on these two electrodes throughout the potential window of interest; however, the selectivity of products is quite different (Fig. S21–S23). Cu-Ga catalyst favors the formate generation, delivering a maximum formate FE of ~38.4% at -1.05 $V_{RHE}$, which is 4 times as high as that on bare Cu. At more negative potential regime, this formate selectivity is gradually taken over by $C_{1+}$ hydrocarbons and oxygenates. This trend of increasing $C_{1+}$ productivity as potential decreases was also observed in most of Cu-based alloys and pure Cu[39].

By conducting a CN (Fig. 5b) and composition-dependent (Fig. 5c) analysis at −1.4 $V_{RHE}$, we confirmed that $C_{1+}$ productivity of Cu-Pd increases at low CN and moderate Cu content, and formate productivity increases at high CN and low Cu content. It is important to note that the compositions of the alloys depicted in Fig. 5c were calculated by considering only the atoms present in active motifs. To validate whether the compositions of these active motifs represent the compositions of the bulk structures, we created a parity plot comparing the two, which revealed a correlation between them (Fig. S24). We also calculated the productivity dependent on bulk composition and compared these values with those dependent on active motif composition (Fig. S25). The overall trends were in agreement,

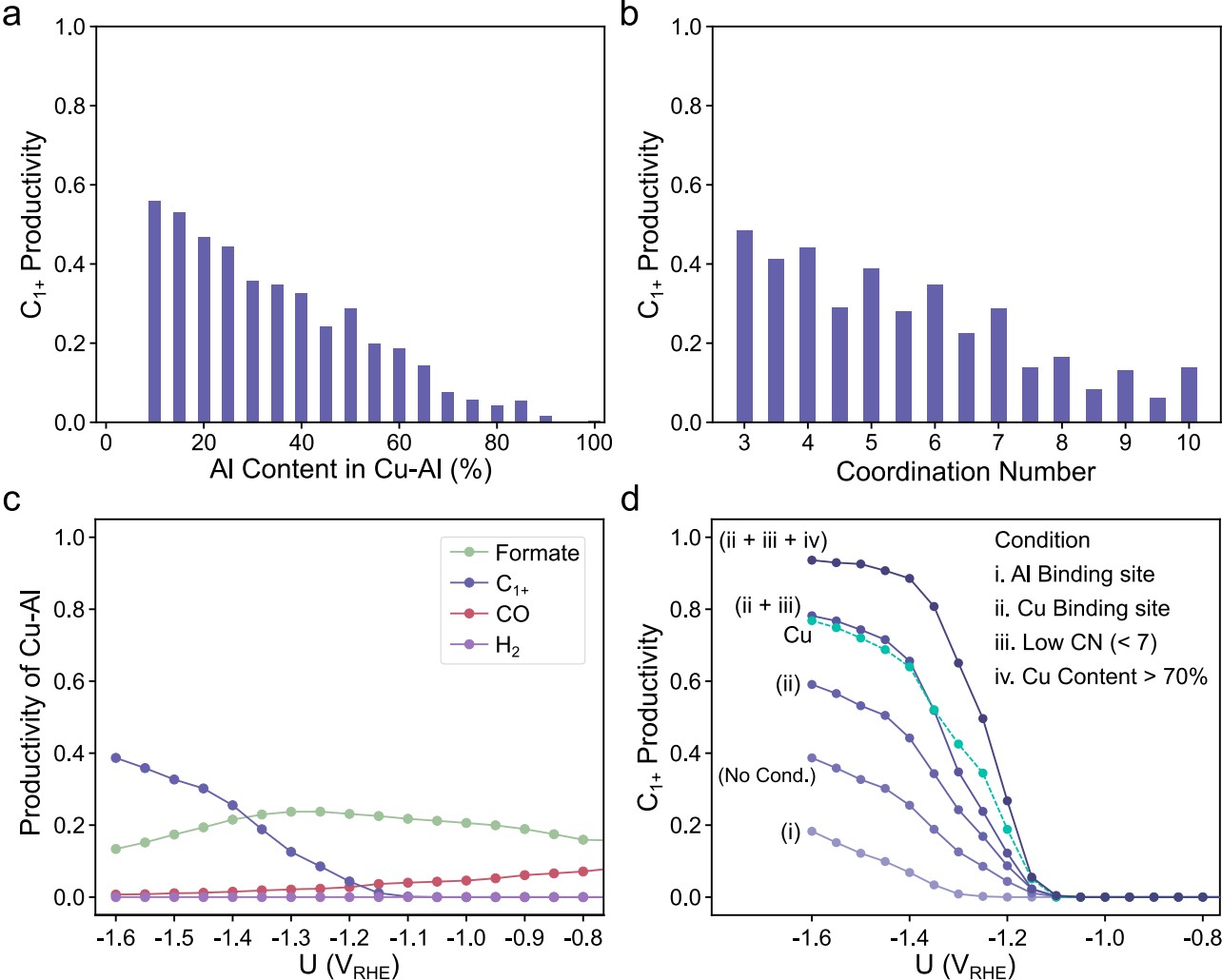

**Fig. 4 | Coordination and composition-based analysis.** $C_{1+}$ productivity of Cu-Al binary alloys with respect to (**a**) Al contents in active motifs and (**b**) coordination numbers (CN) at −1.4 $V_{RHE}$. CN is calculated by averaging the number of the first nearest neighbors of each active site atom. The first nearest neighbors were identified using Voronoi algorithms implemented in Pymatgen[60]. **c** Productivity of Cu-Al without any applied constraints. **d** $C_{1+}$ productivity of Cu-Al catalysts when different conditions of active motifs are applied; (i-ii) active site elements, (iii) CN and (iv) composition. Cu (turquoise dashed line) is also plotted for comparison. All productivities of pure Cu and masked Cu-Al can be found in Fig. S11.

suggesting that the composition of active motifs can indeed serve as a representative approximation of bulk compositions.

**Experimental validations of Cu-Pd catalysts**

Based on the prediction by ML, we experimentally validate the $CO_2$-to-$C_{1+}$ performance of Cu-Pd catalyst. We note in passing that, given that several factors influence a material's synthesizability, it is possible that the proposed crystal structures may not be realized in practice. While our method can suggest more reliable elemental combinations to achieve the desired catalytic activity and selectivity, it does not guarantee the formation of stable alloys. Thus, our method should be used as a tool for prioritizing candidate combinations. The binary electrode was prepared by a galvanic displacement reaction between Cu and Pd(II) species ("Methods"). Typical SEM images in Fig. S26 depict the morphology of Cu-Pd prior to and post $CO_2RR$ electrolysis. In contrast to the relatively flat surface of polished Cu (Fig. S15), the Cu-Pd electrode consists of densely packed nanoparticles, which contributes to the ~ 12-times higher surface roughness compared to bare Cu (Fig. S27). Figure S28 plots the depth profile of Cu-Pd electrode as extracted from ex situ time of flight secondary-ion mass spectrometry (TOF-SIMS), in which a surface enrichment of Pd is clearly observed in

the re-constructed 3D plots shown in Fig. 6a, and in good agreement with the determined near-surface composition, i.e., an atomic ratio of 4:1 for Cu: Pd, from XPS results (Fig. S29 and Table S4). Grazing-incidence X-ray diffraction with Rietveld refinement has been carried out to further probe the crystalline structure of Cu-Pd electrode at an incidence angle of 0.5°. As shown in Fig. S30, the near-surface layer consists mainly of the metallic Cu component with ~1.3% $Cu_2O$ phase. No Cu-Pd alloy was detected, likely due to the highly dispersed feature of Pd decoration and its overall low doping content of 1.0−1.2 at.% from the bulky EDS analysis (Fig. S31).

The electrochemical $CO_2RR$ performance of Cu-Pd electrode with reference to polished Cu was then evaluated by chronoamperometric electrolysis from -0.75 to -1.15 $V_{RHE}$. In the potential range studied, a much higher overall current density was observed on Cu-Pd compared to Cu (Fig. 6b), which is associated with the increased surface roughness. Moreover, as shown in Fig. 6c, a suppressed $H_2$ FE was found on Cu-Pd throughout the potential window studied, suggesting the main contribution from $CO_2RR$ rather than HER to the current density enhancement. As shown in Figs. S32, S33, the major $C_1$ products on polished Cu includes CO (max. FE of 12.8% at −0.85 V), formate (max. FE of 23.7% at -0.95 V), and $CH_4$ (max. FE of 33.5% at −1.15 V), where a

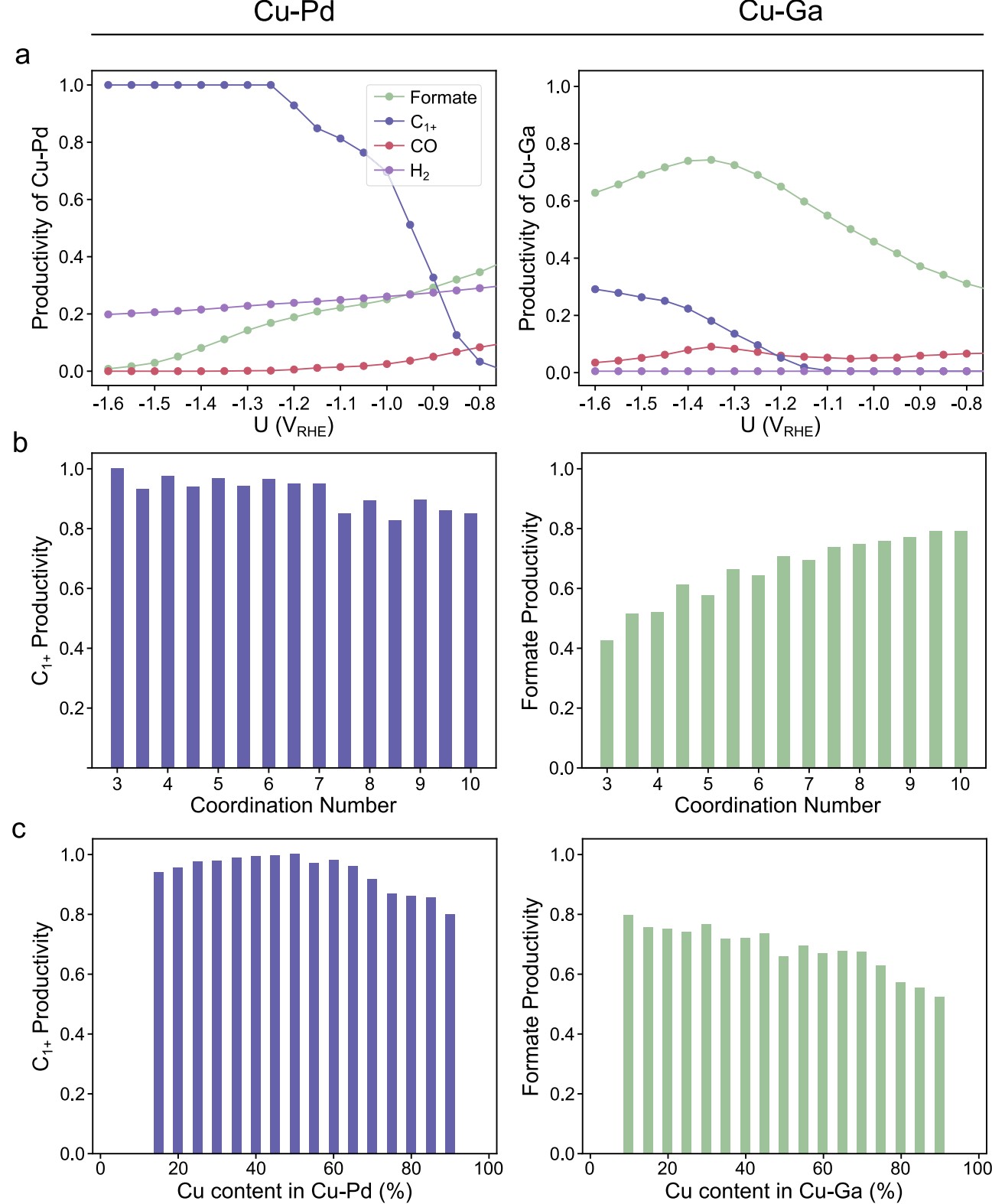

**Fig. 5 | Prediction results of Cu-Pd and Cu-Ga. a** Productivities of Cu-Pd (left) and Cu-Ga (right) binary alloys as a function of the applied potentials, where ML-predicted binding energies were used. **b** Coordination number and (**c**) composition-dependent $C_{1+}$ and formate productivity of Cu-Pd and Cu-Ga at −1.4 $V_{RHE}$, respectively.

peak $C_{2+}$ products selectivity of 38.2% locates at −1.05 V. As to Cu-Pd, the C-C coupling selectivity was dramatically improved, i.e., $C_2H_4$ as representative hydrocarbon product and ethanol as representative oxygenate emerged at an onset potential of −0.75 V and continued to increase with negative-going potential (Fig. S33). $C_2H_4$ showed the

highest selectivity of 32.3% FE at −1.15 V (Fig. 6d), which is ca. 3.6 times as high as that on bare Cu. Moreover, Fig. 6e shows the fraction of CO produced by the $CO_2RR$ converted to $C_{1+}$ products (including $CH_4$ and $C_{2+}$) rapidly increases on Cu-Pd from 0.29 at -0.75 V to 0.84 at −0.85 V, and finally reaches 0.99 at −1.15 V. Noteworthy, in the $CO_g$

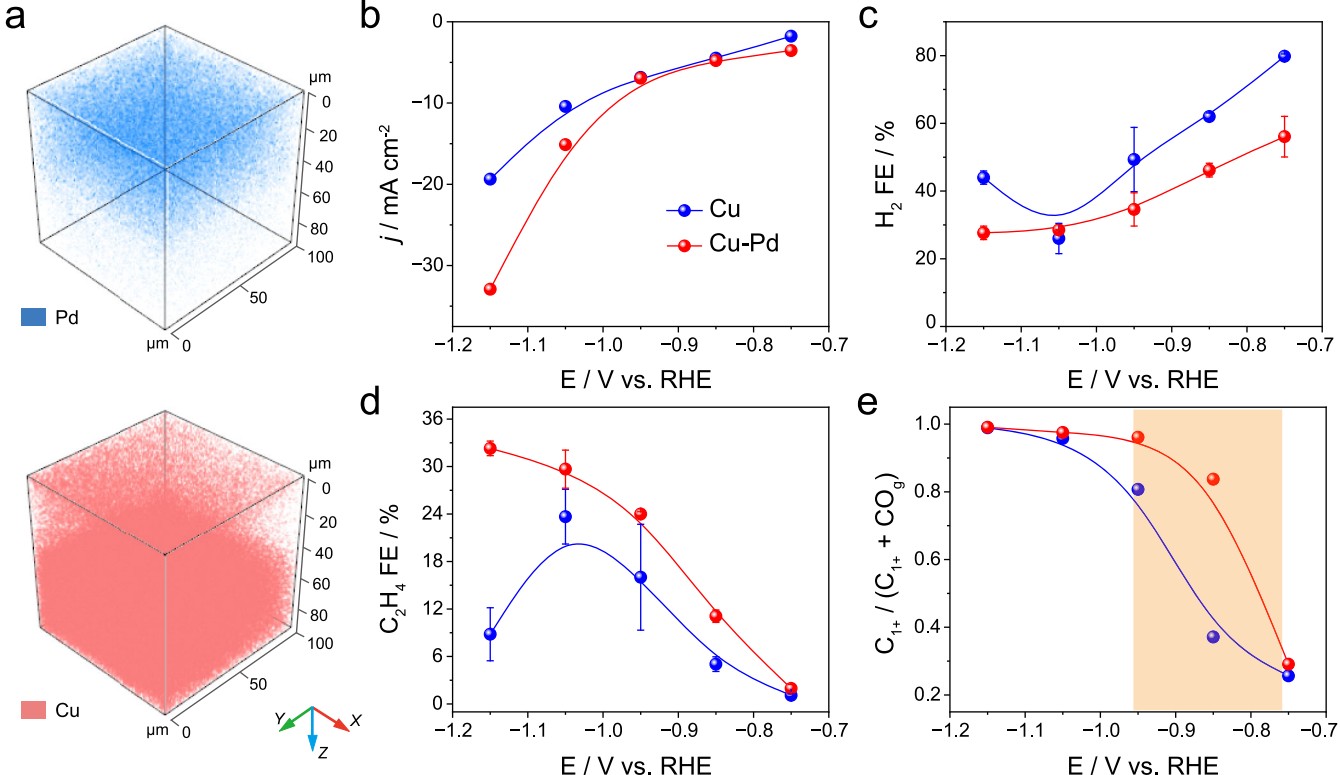

**Fig. 6 | Experimental results of Cu-Pd. a** Reconstructed 3D plots for Cu-Pd electrode based on the depth profile of TOF-SIMS spectra. **b** Steady-state current densities, (**c**) the Faradaic efficiencies of $H_2$, (**d**) $C_2H_4$ and (**e**) the calculated ratio of $C_{1+}$ products to the sum of $CO_g + C_{1+}$ products on polished Cu versus plated Cu-Pd electrodes under different applied potentials.

accumulation regime around −0.85 V (marked in orange shadow), a 2.3 times enhancement on the $C_{1+}$ selectivity is observed on Cu-Pd compared to bare Cu electrode, successfully validating the above theoretical predictions of predominant $C_{1+}$ productivity.

## Discussion

In this work, we combined active motif-based binding energy prediction model, DSTAR, with the advanced 3D $CO_2RR$ selectivity map to perform HTVS. Although our strategy has proved its validity and potential, there is room for improvement in addition to improving prediction accuracy of ML models.

Since DSTAR pipeline generates active motifs by simply substituting elements, it is necessary to evaluate thermodynamic stability and synthesizability of corresponding bulk structures to accurately simulate experimental conditions. However, DFT calculations to validate stability of all candidate bulk structures can be computationally intensive, which may offset the benefits of ML-assisted HTVS, e.g., cost-effective exploration of large chemical space. To address this issue, integrating models predicting stability of unrelaxed crystal structures with high accuracy could be a solution[40–42].

Additionally, the selectivity map used in our workflow was constructed based only on thermodynamic conditions, not considering additional factors such as kinetics, phase transitions and local field effects. This might result in the discrepancy observed in Ag, Pd and Zn cases as discussed above. We expect that it could be addressed by introducing more advanced methods, such as constant-potential calculations of electrochemical reactions[43,44], or more advanced descriptor. For example, Ringe recently reported that potential of zero charge can be used as the second descriptor in addition to binding energy, which can more accurately predict the experimental results[45]. Moreover, recent theoretical studies, utilizing constant-potential calculations, have suggested that the CO selectivity of certain catalysts is

related to the kinetics of OCO* binding and its subsequent protonation[33,46]. The development of new descriptors that encompass various aspects of electrocatalysis is crucial for enhancing the validity of HTVS, and will be a focus of future research.

In summary, we developed ML-assisted HTVS strategy, which predicts catalytic activity and selectivity of $CO_2RR$ for the expanded materials chemical space. Because the ML inputs include information on composition and coordination number, this approach not only identifies promising elemental combinations but also provides a more detailed catalyst design strategy, which is one step forward to the current HTVS approach. Using this strategy, we predicted the dominant product, and both its activity and selectivity for 465 elemental combinations. Among them, we selected Cu-Pd which has the highest productivity for $C_{1+}$, and Cu-Ga binary alloys which has barely been reported in literature. Further experimental validations indeed confirmed its high intrinsic selectivity toward $C_{1+}$ and formate, respectively. We expect our strategy developed in this work to accelerate the discovery of active and selective $CO_2RR$ catalysts.

## Methods
### Calculation details
We performed spin-polarized density functional theory (DFT) calculations using Vienna Ab initio Simulation Package (VASP, version 5.4.4) code[47,48] with the projector augmented wave (PAW) pseudopotential method[49] and the generalized gradient approximation- revised Perdew-Burke-Ernzerhof (GGA-RPBE) exchange-correlation functional[50]. For geometry optimizations, energy and force criteria for the convergence were set to $10^{-5}$ eV and 0.03 eV/Å, respectively, and the kinetic energy cutoff was set to 500 eV. Monkhorst-Pack k-point mesh was set to $(k_1 \times k_2 \times 1)$ to satisfy 25 Å $< a_n \times k_n$ $(n = 1, 2) < 30$ Å, where $a_1$ and $a_2$ are the sizes of unit vectors in $x$ and $y$ directions, respectively[51]. As reference data of pure metals, (111) and (211) facets of

face-centered cubic (FCC) metals (Ag, Au, Cu, Ir, Ni, Pb, Pd, Pt, Rh) were modeled using three-layered (3 × 3) cell.

The Gibbs free energies of reactions were calculated using computational hydrogen electrode (CHE) method. This method assumes the equivalent chemical potential of 0.5 $H_2$ and a pair of proton and electron ($H^+ + e^-$) at 0 $V_{RHE}$ (Reversible Hydrogen Electrode) and the standard conditions[52]. The effect of the applied potential ($U$) was included as $G(H^+ + e^-) = 0.5 G(H_2) - eU$. The Gibbs free energy correction ($G_{corr}$) was used to convert DFT energies into the Gibbs free energy following $G_{corr} = E_{zpe} + \int C_p \, dT - TS + E_{solv} + E_{gas}$, where $E_{zpe}$, $\int C_p \, dT$, and $-TS$ are zero-point energies, enthalpic and entropic contributions, respectively. $E_{solv}$ is a solvation correction for adsorbates[39] and $E_{gas}$ is a gas-phase correction of GGA-RPBE functional[53]. The correction values were calculated using the ideal gas and the harmonic oscillator approximation for gaseous molecules and adsorbates, respectively, as implemented in Atomic Simulation Environment (ASE)[54]. The correction values for gaseous molecules and adsorbates are summarized in Table S5 and S6, respectively.

## Machine learning details
To predict binding energies of adsorbates (CO*, H*, OH*) on a wide variety of active motifs, we used DSTAR (DFT & Structure-free Active motif based Representation) method which converts active motifs into machine learning inputs[7]. The inputs consist of 36 components, where weighted average values of various elemental properties in three categorized sites (FNN, $SNN_{same}$, $SNN_{sub}$) and the number of atoms in each site are concatenated. Note FNN, $SNN_{same}$ and $SNN_{sub}$ correspond to the first nearest neighboring atoms of the binding site, the second nearest neighboring atoms of the binding site in the same layer, and the sublayer, respectively. The concatenated inputs are standardized by applying StandardScaler method as implemented in scikit learn[55]. An example of fingerprint generation is illustrated in Fig. S1.

Gradient boosting regressor[56] and XGBoost regressors[57] were chosen for CO*/H* and OH* binding energy predictions, respectively, as they achieved the best prediction accuracy. The hyperparameters of each model were determined by Bayesian optimization (Table S7). 16,097, 6512 and 18,362 data of CO*, OH* and H* adsorptions taken from GASpy dataset[9] were used to train/test ML models, respectively. For ML model training, we performed fivefold cross validations with 80% train and 20% test sets. To predict binding energies of newly generated active motifs, we used ML models trained with the whole data.

## Productivity calculation
For the given data point on the 3D selectivity map, we convert the predicted point into a cuboid volume, where the twice prediction errors of $\Delta E_{H^*}$, $\Delta E_{OH^*}$ and $\Delta E_{CO^*}$ correspond to the length, width and height of the cuboid, respectively. These cuboid volumes, which partially occupy the product region, are utilized to calculate the productivity, where the contribution of each volume to each product is calculated considering its energetics. From the Gibbs free energy diagram toward certain products, we calculated the maximum reaction barrier ($\Delta G_{MAX}$) at the applied potential as

$$\Delta G_{max} = \max_i (\Delta G_{rxn}^i) \qquad (1)$$

where $\Delta G_{rxn}^i$ is a reaction Gibbs energy of an elementary step $i$ in the reaction pathway (Fig. S34). Only $\Delta G_{MAX} < 0$ eV is further considered, which corresponds to the spontaneous thermodynamics. To ensure that points with lower $\Delta G_{MAX}$ have a higher contribution to the overall activity, a weighted $\Delta G_{MAX}$ value was obtained as follows:

$$\Delta w G_{max} = e^{-\Delta G_{max}} \qquad (2)$$

Eventually, the productivity ($p_k$) of a specific elemental combination for a given product $k$ (formate, CO, $C_{1+}$ and $H_2$) is determined by

summing over $\Delta w G_{max,i,k}$ multiplied by $v_{i,k}$.

$$p_k = \frac{\sum_i^N \Delta w G_{max,i,k} * v_{i,k}}{N} \, for \, \Delta w G_{max,i,k} > 1 \qquad (3)$$

Here, $v_{i,k}$ represents the partial volume of the cuboid for the $i$ th active motif toward product $k$. This accounts for potential issues stemming from the uncertainty in ML predictions and the discontinuity of the boundary conditions, which could result in incorrect predictions (Fig. S35). N represents the total number of unique active motifs, which remains identical within the monometallic system and the bimetallic system. After calculating the productivities, they were normalized using the MinMaxScaler across all elemental combinations..

## Electrode preparation
The Cu foil (0.1 mm thick, 99.9999%, Alfa Aesar) working electrodes were first cleaned by sonication in acetone, ethanol and deionized water for 2 min, respectively. Then the Cu foil was electrochemically polished in 85% $H_3PO_4$ by applying a voltage of +3 V versus carbon paper for 180 s. The Cu-Ga electrode preparation and relevant characterizations can be found in Supplementary Note 3.

To prepare the Cu-Pd electrode, the electropolished Cu foil was soaked in the solution containing 0.5 mM palladium sodium chloride for 30 min, where the galvanic displacement reaction between Cu and Pd(II) species may occur. Then the as-prepared Cu-Pd electrode was rinsed with deionized water and dried by nitrogen flow.

## Electrode characterizations
The surface morphologies of the Cu-Pd films before and after electrolysis were characterized by a scanning electronic microscopy (SEM, Sigma 300) with equipped energy-dispersive X-ray spectroscopy (EDS) detector. The surface electronic structures of the plated Cu-Pd films were analyzed by using ESCALAB 250 XI X-ray photoelectron spectrometer (Thermo Scientific) with the monochromatic Al Kα radiation (1486.6 eV), and the binding energies were calibrated with reference to C1s peak at 284.8 eV. Contact angle measurements were performed with a JY-82B Kruss DSA instrument. To identify the crystalline phases of surface films, grazing-incidence X-ray diffraction (GIXRD) patterns of CuPd electrodes from 30 to 65° were recorded at a grazing angle of 0.5° on Bruker D8 Discover spectrometer. Time-of-flight secondary-ion mass spectrometric measurements were run on ION-TOF TOF-SIMS 5, with the pressure of analysis chamber below $1.1 \times 10^{-9}$ mbar and the pulsed $Bi^{3+}$ ion beam of 30 keV for high mass resolution analysis. 30 keV GCIB+ ion beam sputtering and a 300 × 300 μm² sputter raster were deployed for depth profile study.

## Electrochemical measurements
All electrochemical measurements were run at 25 °C in a customized gastight H-type glass cell separated by Nafion 117 membrane[58,59]. The plated Cu-Pd, Cu-Ga film electrode or electrochemically polished Cu foil was deployed as the working electrode, a graphite rod (99.995%, Aldrich) and an Ag/AgCl electrode were used as the counter electrode and the reference electrode, respectively. 0.05 M $Cs_2CO_3$ (99.99%, Adamas-Beta) dissolved in Milli-Q water was used as the electrolyte, which was further purified by electrolysis between two graphite rods at 0.1 mA for 24 h to remove trace amount of metal ion impurities. Prior to CO2RR electrolysis, 50 sccm $CO_2$ (Air Liquid, 99.999%) was bubbled for at least 30 min to get the 0.1 M $CO_2$-saturated $CsHCO_3$ electrolyte, and a constant $CO_2$ gas flow (30 sscm, monitored by Alicat mass flow controller) was continuously delivered into the cathodic compartment to keep $CO_2$-saturation during electrolysis.

Electrochemical responses were recorded on a Biologic VSP-300 potentiostat. The solution resistance ($R_u$) was determined by potentiostatic electrochemical impedance spectroscopy (PEIS) at

frequencies ranging from 0.1 Hz to 200 kHz, and manually compensated as $E$ ($iR$-corrected vs. RHE) = $E$ (vs. RHE) - $R_u \times i$ (amps of averaged current), with a manual $i$R drop compensation at 80% level. All potentials (if not specifically mentioned) in this work were converted to the RHE scale as $E$ (vs. RHE) = $E$ (vs. Ag/AgCl) + 0.197 V + 0.0591 × $pH_{bulk}$. The electrochemical active surface area (ECSA) of Cu-based electrodes was estimated via its double layer capacitance by measuring and plotting the double layer charging current as a function of scan rate.

## Quantification of CO₂RR products

The gas products from the electrochemical cell was analyzed by a Shimadzu 2014 gas chromatography (GC) equipped with a thermal conductivity detector (TCD) for $H_2$ concentration quantification and a flame ionization detector (FID) coupled with a methanizer for quantifying hydrocarbons concentration. UHP Ar was used as the carrier gas and constituents of the gaseous sample were separated using two Porapak N80/100 columns packed with Molecular Sieve-13X. Faradaic efficiency (FE) of certain reduction product was calculated as:

$$FE_i = \frac{x_i \nu nF}{V \times j} \times 100\%$$

where $x_i$ is the volume fraction of specie $i$ as determined by on-line GC, $\nu$ is the flow rate, generally set at 30 sccm being monitored by an Alicat mass flow controller, $n$ is the electron transfer number, $F$ is the Faradaic constant, $V$ is the molar volume of ideal gas under CO₂RR operation condition, $j$ is the total current density.

The aqueous products were analyzed by a 500 MHz nuclear magnetic resonance (NMR) spectrometer from Bruker Company with a water suppression technique. Typically, 450 μL of the electrolyte after 5400-s electrolysis was mixed with 50 μL of $D_2O$ solution containing DMSO as the internal standard.

## Data availability

The binding energy of CO*, OH*, H* and corresponding surface structure can be found in https://github.com/ulissigroup/GASpy or https://github.com/SeoinBack/DSTAR-CO2RR.

## Code availability

The Python code of DSTAR used in this study can be found in https://github.com/SeoinBack/DSTAR-CO2RR.

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

## Acknowledgements

S.B. acknowledges the support from the National Research Foundation of Korea (NRF) funded by the Ministry of Science and ICT (2015M3D3A1A01064929) and the Carbon Neutral Industrial Strategic Technology Development Program (RS-2023-00261088) funded by the Ministry of Trade, Industry & Energy (MOTIE, Korea). K.J. acknowledges the National Key R&D Program of China (2022YFB4102000, 2022YFA1505100), the NSFC (22002088), the Shanghai Sailing Program (20YF1420500), and the Science and Technology Commission of Shanghai Municipality (22dz1205500). A generous supercomputing time from KISTI is also acknowledged.

## Author contributions

D.H.M and H.L. contributed equally to this work. D.H.M. and S.B. conceptualized the project. S.B. and K.J. supervised the project. D.H.M. conducted DFT calculations and constructed HTVS pipeline. H.L. and G.Z. performed experimental validation. D.H.M., C.L. and S.B. discussed the results. All the authors wrote the manuscript.

## Competing interests

The authors declare no competing interests.

## Additional information

**Peer review information** : *Nature Communications* thanks the anonymous, reviewers for their contribution to the peer review of this work. A peer review file is available.

