## [Peer Review File · Nature Communications]

Editorial Note: This manuscript has been previously reviewed at another journal that is not operating a transparent peer review scheme. This document only contains reviewer comments and rebuttal letters for versions considered at *Nature Communications* .

REVIEWER COMMENTS

Reviewer #1 (Remarks to the Author):

I thank the authors for carefully going through the raised points and correcting them. In detail

- 1) Has been answered satisfactorily. Due to the changes this is now more convincing.
- 2) The provided github link is helpful for reproducibility and increases the understanding of the paper.
- 3) I appreciate that the authors already determined the prediction error. With this in mind, I would suggest to include error bars in all the figures of the manuscript which could highly increase the trustability of the results.
- 4) Ok.
- 5) Ok.
- 6) Looks great, I think it is good that the authors found another candidate. I would appreciate only if the authors can add one sentence about the matching of a theoretically predicted structure with an experimentally synthesized one. How difficult is this? What are the challenges? Is this always expected to work out perfectly? This will be important for the community who is going to use the predicted selective systems to reproduce them experimentally.
- 7) Ok.
- 8)
 - I do not quite understand why the productivity of a particular configuration should be scaled with the volume which corresponds to the error. This would mean that more erroneous predictions are contributing more? How does this make sense?
 - The equations should be all labeled with numbers
 - The equation $p_k = \dots$ the sum introduced sums over what? Should be explained.
- 9) Ok.
- 10) Convincing.

11) Ok.

Reviewer #2 (Remarks to the Author):

In the present manuscript entitled "From data to discovery: Machine learning enables the identification of active and selective CO₂ reduction electrocatalysts", the authors have established novel high-throughput virtual screening strategy to suggest active and selective catalysts for CO₂RR without being limited to a database.

In the revised manuscript, the authors have done a substantial revision to the previously submitted version, which I highly appreciate. Therefore, I recommend the acceptance of this manuscript for publication in Nature Communications.

Reviewer #3 (Remarks to the Author):

The revised manuscript included discussion about the model details, uncertainty quantification, and further validation of Cu/Pd catalysts. Most of the concerns were addressed.

1. CuGa stability might be a concern. Authors claimed that the CuGa is in metal state, but the bulk pourbiax diagram with an arbitrary 0.5 eV threshold is not sufficient. Is there dissolution or Ga oxide formation in reaction conditions?
2. In terms of referencing to prior work, particularly by Sargent Cu-Al, the selectivity to CO₂ reduction was considered.
3. For the graph models, authors were indicating that the optimized structures are used which is not true. Authors need to revise and accurately reflect the state of the art and their benchmarks.

Reviewer #1 (Remarks to the Author)

I thank the authors for carefully going through the raised points and correcting them. In detail

- 1) Has been answered satisfactorily. Due to the changes this is now more convincing.
- 2) The provided github link is helpful for reproducibility and increases the understanding of the paper.
- 3) I appreciate that the authors already determined the prediction error. With this in mind, I would suggest to include error bars in all the figures of the manuscript which could highly increase the trustability of the results.

We appreciate for your comment. Below, we have compared two versions of parity plots, one with error bars and the other with density-based color gradients. The one with error bars is less visible than the latter, thus we decided to use the latter in the manuscript (**Figure R1**). Furthermore, the productivity figures already account for prediction uncertainty through the productivity equations. You can find more details in the answer to the 8th comment, which is related to the uncertainty of our method.

Figure R1. Parity plots with points and error bars (left) and with only hexagonal density-based color gradients (right).

4) Ok.

5) Ok.

6) Looks great, I think it is good that the authors found another candidate. I would appreciate only if the authors can add one sentence about the matching of a theoretically predicted structure with an experimentally synthesized one. How difficult is this? What are the challenges? Is this always expected to work out perfectly? This will be important for the community who is going to use the predicted selective systems to reproduce them experimentally.

We appreciate this valuable comment. The primary challenge in data-driven catalyst discovery using DFT and ML is the synthesizability of the proposed catalysts in specific crystal structures. Since several factors

influence a material's synthesizability, it is possible that the proposed crystal structures may not be realized in practice. While our method can suggest more reliable elemental combinations, it does not guarantee the formation of stable alloys. Thus, our method should be used as a tool for prioritizing candidate combinations. We added this perspective as follows:

Line 317- 321 in Page 18:

We note in passing that, given that several factors influence a material's synthesizability, it is possible that the proposed crystal structures may not be realized in practice. While our method can suggest more reliable elemental combinations to achieve the desired catalytic activity and selectivity, it does not guarantee the formation of stable alloys. Thus, our method should be used as a tool for prioritizing candidate combinations.

7) Ok.

8)

- I do not quite understand why the productivity of a particular configuration should be scaled with the volume which corresponds to the error. This would mean that more erroneous predictions are contributing more? How does this make sense?

Figure S35. An example of uncertainty contribution for a point near the boundary condition in the 2D projections (right) of 3D selectivity map (left). The uncertainty caused by the ML prediction is represented as a volume based on the MAE of each ML model. Since the volume intersects with the boundary condition, the active motif also contributes to the productivity of the product from another pathway (COOH*) along with the predicted pathway (HCOO*).

As pointed out by the reviewer in the previous review, we acknowledge that the uncertainty of the ML predictions and the discontinuity of the boundary conditions could lead to incorrect predictions, especially

when the predicted points are near the boundary for different selectivities. For instance, the white point representing predicted binding energies in **Figure S35** would be identified as selective for the HCOO* pathway. However, considering the ML errors, there is a possibility that the actual energies for that point are selective for the COOH* pathway.

To address this issue, we represented each predicted point (i) as a 3D volume ($v_{i,k}$), where the intersected volume contributes to each reaction pathway leading to product (k). In **Figure S35**, 80 % of the volume is occupied by the HCOO*-selective region, allowing its contribution to be calculated.

- The equations should be all labeled with numbers

We labeled the equations with numbers in method section.

- The equation $p_k = \dots$ the sum introduced sums over what? Should be explained.

For clarity, we have added further details as copied below.

Line 448-452 in Page 25:

For the given data point on the 3D selectivity map, we convert the predicted point into a cuboid volume, where the twice prediction errors of ΔE_{H^*} , ΔE_{OH^*} and ΔE_{CO^*} correspond to the length, width and height of the cuboid, respectively. These cuboid volumes, which partially occupy the product region, are utilized to calculate the productivity, where the contribution of each volume to each product is calculated considering its energetics. From the Gibbs free energy diagram toward certain products, we calculated the maximum reaction barrier (ΔG_{MAX}) at the applied potential as

$$\Delta G_{MAX} = \max_i(\Delta G_{rxn}^i) \quad (1)$$

, where ΔG_{rxn}^i is a reaction Gibbs energy of an elementary step i in the reaction pathway (**Figure S34**).

Only $\Delta G_{MAX} < 0$ eV is further considered, which corresponds to the spontaneous thermodynamics. To ensure that points with lower ΔG_{MAX} have a higher contribution to the overall activity, a weighted ΔG_{MAX} value was obtained as follows:

$$\Delta wG_{max} = e^{-\Delta G_{max}} \quad (2)$$

Eventually, the productivity (p_k) of a specific elemental combination for a given product k (formate, CO, C₁₊ and H₂) is determined by summing over $\Delta wG_{max,i,k}$ multiplied by $v_{i,k}$.

$$p_k = \frac{\sum_i^N \Delta wG_{max,i,k} * v_{i,k}}{N} \text{ for } \Delta wG_{max,i,k} > 1 \quad (3)$$

Here, $v_{i,k}$ represents the partial volume of the cuboid for the i th active motif toward product k . This accounts for potential issues stemming from the uncertainty in ML predictions and the discontinuity of the boundary conditions, which could result in incorrect predictions (**Figure S35**). N represents the total

number of unique active motifs, which remains identical within the monometallic system and the bimetallic system. After calculating the productivities, they were normalized using the MinMaxScaler across all elemental combinations.

9) Ok.

10) Convincing.

11) Ok.

Reviewer #2 (Remarks to the Author):

In the present manuscript entitled “From data to discovery: Machine learning enables the identification of active and selective CO₂ reduction electrocatalysts”, the authors have established novel high-throughput virtual screening strategy to suggest active and selective catalysts for CO₂RR without being limited to a database.

In the revised manuscript, the authors have done a substantial revision to the previously submitted version, which I highly appreciate. Therefore, I recommend the acceptance of this manuscript for publication in Nature Communications.

Reviewer #3 (Remarks to the Author):

The revised manuscript included discussion about the model details, uncertainty quantification, and further validation of Cu/Pd catalysts. Most of the concerns were addressed.

1. CuGa stability might be a concern. Authors claimed that the CuGa is in metal state, but the bulk pourbiax diagram with an arbitrary 0.5 eV threshold is not sufficient. Is there dissolution or Ga oxide formation in reaction conditions?

Figure R2. (a-b) The calculated 2D Pourbaix diagrams of (a) GaCu₃ (mp-1183995) and (b) Ga₄Cu₉ (mp-1197621). The color bar indicates the Pourbaix decomposition energy (ΔG_{pbx}) of each compound, where a lower ΔG_{pbx} corresponds to higher stability. The compounds shown in the diagrams represent the stable phases at the given potential and pH. (c-d) The 1D Pourbaix diagrams of (c) GaCu₃ and (d) Ga₄Cu₉ at pH 8.6. The red solid lines are ΔG_{pbx} of each compound as a function of the potential, and the areas under the lines indicate the most stable phase within that potential range. The gray dashed horizontal line indicates the ΔG_{pbx} criteria for metastable phases (10.1021/acs.chemmater.7b03980). The experimental validation was performed from -0.75 to -1.15 V_{RHE} (green area).

We apologize for not providing a sufficient explanation of the Pourbaix diagram. To help understanding, we have included a 2D Pourbaix diagram and additional details in the figure (**Figure R1**).

Both the 1D and 2D Pourbaix diagrams indicate that GaCu₃ and Ga₄Cu₉ become metastable with a stability of under 0.5 eV/atom after reaching -0.5 V_{SHE}. Furthermore, from potentials ranging from -0.8 V_{SHE} to -1.5 V_{SHE}, the Pourbaix stability values of CuGa alloys (Ga₄Cu₉, GaCu₃ and Ga₂Cu) are nearly 0 eV/atom, signifying their status as the most stable phases.

2. In terms of referencing to prior work, particularly by Sargent Cu-Al, the selectivity to CO₂ reduction was considered.

In prior work by Sargent, CO₂ reduction selectivity was determined based on the H* binding energy, where a weak H* binding energy inhibits the competitive hydrogen evolution reaction, thus increasing selectivity towards CO₂RR.

In our method, we have also accounted for selectivity using boundary conditions. The competition between HER and CO₂RR was considered through boundary condition 5, which determines whether CH* protonation for CO₂RR or surface protonation for HER is more favorable. Additionally, using the boundary conditions, we have explored the selectivity between various CO₂RR products, which was not addressed in Sargent's work.

3. For the graph models, authors were indicating that the optimized structures are used which is not true. Authors need to revise and accurately reflect the state of the art and their benchmarks.

We appreciate your comment. The graph-based approach mentioned in the main text originally referred to the corresponding author's previous method (*The Journal of Physical Chemistry Letters* 10 (15), 4401-4408). However, this method is no longer considered state of the art. Therefore, we have made changes to the sentence, removing the comparison part, as the current mainstream focus is on structure-to-energy/force (S2EF) rather than predicting binding energies.

Line 99-102 in Page 6:

The active motif representation in DSTAR does not require time-consuming steps such as slab structure generations, binding site identifications and iterative optimizations, allowing for the exploration of a wide chemical space, thus facilitating a discovery of novel catalysts.

REVIEWERS' COMMENTS

Reviewer #1 (Remarks to the Author):

All open points have been answered. I recommend the manuscript for publication.

Reviewer #3 (Remarks to the Author):

Authors addressed the comments very well.